# FROM SCORE DISTRIBUTIONS TO BALANCE: PLUG-AND-PLAY MIXTURE-OF-EXPERTS ROUTING

## ABSTRACT

Mixture-of-Experts (MoE) models can scale parameter capacity by routing each token to a subset of experts through a learned gate function. While conditional routing reduces training costs, it shifts the burden on inference memory: expert parameters and activations consume memory, limiting the number of experts per device. As tokens are routed, some experts become overloaded while others are underutilized. Because experts are mapped to GPUs, this imbalance translates directly into degraded system performance in terms of latency, throughput, and cost. We present LASER, a plug-and-play, inference-time routing algorithm that balances load while preserving accuracy. LASER adapts to the shape of the gate's score distribution. When scores provide a clear preference, it routes to the strongest experts; when scores are more uniform, it broadens the set of viable experts and routes to the least-loaded among them. Because LASER relies only on gate scores from a trained model, it integrates directly into existing MoE inference pipelines without retraining or finetuning. We evaluate LASER on Mixtral-8×7B and DeepSeek-MoE-16b-chat across four datasets (ARC-Easy, ARC-Challenge, MMLU, and GSM8K). LASER improves load balancing, translating into lower latency and higher throughput, while keeping accuracy changes negligible.

## 1 INTRODUCTION

Recent breakthroughs in large language models (LLMs) have been driven by scaling parameter counts, which improves accuracy but greatly increases computational cost. Mixture-of-Experts (MoE) extensions to Transformer models (Vaswani et al., 2017) address this by activating only a subset of parameters for each token, enabling scaling to hundreds of billions of parameters at reduced per-token cost (Liu et al., 2024; Dai et al., 2024; Fedus et al., 2022; Lepikhin et al., 2020).

While MoEs reduce training costs, they shift the burden to inference memory. During inference, a gate function assigns tokens to experts, often creating hot experts that receive many tokens and cold experts that receive few. Because experts are placed on GPUs, this imbalance directly translates into uneven GPU utilization: overloaded experts can increase latency or trigger out-of-memory failures, while underutilized experts leave GPU capacity idle. Since inference proceeds at the pace of the most heavily loaded GPU, imbalance directly increases latency, reduces throughput, and raises costs.

Existing works have addressed load imbalance in MoEs at different stages of the pipeline. Device-level placement methods optimize how experts are distributed across GPUs (Huang et al., 2024), but they operate at coarse granularity and cannot adapt to token-level fluctuations during inference. Training-time balancing techniques such as SIMBAL (Omi et al., 2025) and bias-injection methods (Wang et al., 2024) encourage more uniform expert usage during training. However, even when effective during training, they do not guarantee balanced usage at inference. Moreover, these methods require retraining or modifications to the training loop. This makes them costly and limits their applicability to existing pretrained models.

Routing is central to MoE layers, as it determines which experts process each token. A lightweight gating network produces a score distribution over experts, indicating their relevance to the token. The standard approach, top-k routing (Shazeer et al., 2017), selects the k experts with the highest scores. However, this strategy fixes the choice to the top-k experts, overcommits to a few high-scoring experts, and ignores the overall shape of the score distribution. As a result, it misses opportunities for balancing when the distribution is nearly uniform.

We propose LASER, a plug-and-play inference-time routing algorithm that adapts to the shape of the gate score distribution. Our analysis of gate scores across layers in Mixtral and DeepSeek shows that distributions vary: early and late layers are sharply skewed toward a few experts, whereas middle layers are flatter and spread probability more evenly. LASER exploits this variability by broadening the candidate set when scores are diffuse and narrowing it when scores are sharply peaked. If the top-$k$ experts already dominate the score mass, it follows those experts; otherwise, it forms a candidate pool by thresholding scores to include all plausibly relevant experts. From this pool, LASER assigns tokens to the least-loaded experts, balancing utilization while preserving score quality. Because LASER relies only on gate scores from the trained model, it integrates directly into existing MoE inference pipelines without retraining. While this paper focuses on reducing load imbalance, the same routing framework could be extended to other system-level objectives, such as prioritizing experts on the same node to reduce communication or incorporating memory pressure constraints. This flexibility highlights that LASER is not limited to a single-purpose metric but provides a general mechanism for inference-time routing.

We integrated LASER into two existing MoE models, DeepSeek-MoE and Mixtral, and evaluated them on four datasets: ARC-Easy, ARC-Challenge (Clark et al., 2018), MMLU (Hendrycks et al., 2021), and GSM8K (Cobbe et al., 2021). Our experiments show that LASER reduces expert-load imbalance by up to $1.92\times$ ($\approx 48\%$) and maintains accuracy within 0.02 absolute ($\leq 2\%$) of baseline top-$k$ routing. This paper makes three contributions: (1) We characterize gate score distributions across layers of different MoE models and show that fixed top-$k$ routing ignores this variability.(2) We present LASER, a plug-and-play inference-time algorithm that adapts candidate pools to gate score distributions and routes tokens to the least-loaded experts without retraining. (3) We demonstrate the effectiveness of LASER through experiments on large MoE models, validating its impact on imbalance, latency, and accuracy.

## 2 RELATED WORKS

**MoE Transformers.** Mixture-of-Experts (MoE) models use a gating network to route each token to a small subset of experts, activating only those experts while others remain idle (Shazeer et al., 2017). This sparse activation enables large model capacity with limited per-token computation. The Switch Transformer (Fedus et al., 2022) replaces each feedforward block with several experts but routes to only one expert per token. Mixtral (Jiang et al., 2024) instead uses eight experts per layer and selects two per token. DeepSeekMoE (Dai et al., 2024) introduces fine-grained experts and a small set of shared experts to improve specialization and reduce redundancy. Although primarily focused on training-time design, these approaches complement inference-time load-balancing. When GPU memory is limited, inactive expert weights are offloaded to CPU memory, incurring costly CPU–GPU transfers. Pre-gated MoE (Hwang et al., 2024) addresses this by pre-selecting experts for the next block, overlapping expert migration with current execution. Similarly, SiDA-MoE (Du et al., 2024) employs a data-aware predictor to predict expert activations and proactively offload inactive experts to CPU memory.

**Load Balancing in MoEs.** Load imbalance has been addressed at several stages. *Device-level placement.* Huang et al. (Huang et al., 2024) model expert placement across GPUs as a multi-way partitioning problem and propose heuristics based on historical activation. LASER complements these methods by smoothing load at inference time without moving parameters. *Training-time balancing.* SIMBAL (Omi et al., 2025) adds a regularizer to gating weights to promote uniform usage, while Wang et al. (Wang et al., 2024) injects expert-wise biases into gating scores. Both approaches improve balance but require retraining, unlike LASER, which works at inference without modifying model objectives. *Adaptive routing.* Ada-K (Yue et al., 2024) varies the number of experts per token, assigning more to difficult tokens. In contrast, LASER keeps $k$ fixed but selects experts in a load-aware way to reduce imbalance.

LASER complements these approaches by providing a fine-grained, inference-only solution that dynamically adapts to both score distribution and real-time loads.

## 3 MOTIVATION

We motivate our design in three steps. First, we explain how load imbalance impacts system performance (Section 3.1), showing that the latency, throughput, and cost of MoE inference are directly

governed by the heaviest-loaded expert. Second, we show that despite training-time balancing strategies, experts still receive uneven token loads at inference (Section 3.2). Finally, we take a closer look at per-layer gate score distributions and demonstrate that layers differ systematically in their score distributions (Section 3.3), motivating inference-time algorithms that adapt to this variability.

## 3.1 How Load Imbalance Impacts MoE Inference Performance

In MoE inference, each decoding step routes tokens (via all-to-all) to experts distributed across GPUs. A step for batch $b$ completes only when the slowest routed path in each MoE layer finishes. Consequently, the step time $T_{\text{step}}^{(b)}$ is governed by the most loaded component along that path (e.g., a GPU hosting one or more experts). Load imbalance therefore increases latency, reduces throughput, and raises cost (Fedus et al., 2022; Shazeer et al., 2017).

Modern deployments combine expert placement (packing multiple experts per GPU, replicating hot experts) (Huang et al., 2024) with parallelism strategies (data, tensor/model, pipeline, and expert parallelism) (Shoeybi et al., 2019). Our method operates at routing time, per token and per batch, to smooth short-term load fluctuations that static placement cannot respond to quickly. When multiple experts are colocated, the GPU's instantaneous load is the sum of its resident experts; reducing peak expert load reduces GPU peak and straggling. For replicated experts, we treat each replica as an independent routing target and balance across replicas to limit the maximum. Under expert parallelism, balancing across shards and replicas shortens the critical path even when an expert spans devices. With tensor/model and pipeline parallelism, the slowest stage sets step time; smoothing expert loads reduces the chance that any TP/PP stage becomes a bottleneck (Xu et al., 2021; Hwang et al., 2023).

In what follows, we quantify expert-level imbalance and map it to GPU-level imbalance under a given placement. We report expert imbalance as a deployment-agnostic metric and then translate it to system-level impact per deployment using this mapping. It is natural that other optimizations (e.g., placement/replication/packing, communication overlap) may amplify or reduce the realized gains (Hwang et al., 2023; Rajbhandari et al., 2022; Zhong et al., 2024; Kwon et al., 2023); accordingly, expert to GPU imbalance translation can differ across approaches and configurations.

Let $N_{e,L}^{(b)}$ be the number of tokens assigned to expert $e \in \{1, \ldots, n\}$ in layer $L$ of batch $b$, and let

$$\bar{N}_L^{(b)} = \frac{1}{n} \sum_{e=1}^n N_{e,L}^{(b)}, \qquad I_L^{(b)} = \frac{\max_e N_{e,L}^{(b)}}{\bar{N}_L^{(b)}}, \qquad I_{\text{agg}}^{(b)} = \sum_L w_L I_L^{(b)}.$$

Let $\mathcal{G} = \{1, \ldots, G\}$ index GPUs and $A \in \mathbb{R}_{\geq 0}^{G \times n}$ encode placement: $A_{g,e} > 0$ iff expert $e$ runs (fully or partially) on GPU $g$, with $\sum_g A_{g,e} = 1$ for each $e$ (pure placement: $A_{g,e} \in \{0,1\}$). The induced GPU load in layer $L$ is

$$N_{g,L}^{(b)} = \sum_{e=1}^n A_{g,e} N_{e,L}^{(b)}, \qquad \bar{N}_{L,\text{GPU}}^{(b)} = \frac{1}{G} \sum_{g=1}^G N_{g,L}^{(b)}.$$

We define GPU-level imbalance by

$$I_{L,\text{GPU}}^{(b)} = \frac{\max_g N_{g,L}^{(b)}}{\bar{N}_{L,\text{GPU}}^{(b)}}, \qquad I_{\text{agg,GPU}}^{(b)} = \sum_L w_L I_{L,\text{GPU}}^{(b)}.$$

We model the decoding step time for batch $b$ as $T_{\text{step}}^{(b)} \approx \gamma I_{\text{agg,GPU}}^{(b)} + T_{\text{comm}}^{(b)} + T_{\text{offload}}^{(b)}$, where $\gamma$ captures compute on the critical path, $T_{\text{comm}}^{(b)}$ the all-to-all cost (given parallelism and network), and $T_{\text{offload}}^{(b)}$ CPU↔GPU movement under memory pressure.

Empirically, latency scales nearly linearly with $I_{\text{agg, GPU}}^{(b)}$. When the constant term $C = T_{\text{comm}} + T_{\text{offload}}$ is small relative to the compute term, throughput scales inversely with GPU imbalance; relative to a baseline (named base),

$$\frac{\text{throughput}}{\text{throughput}_{\text{base}}} \approx \frac{I_{\text{agg,GPU}}^{(b,\text{base})}}{I_{\text{agg,GPU}}^{(b)}}.$$

Similarly, average cost per token $\propto T_{\text{tok}}$ scales with $I_{\text{agg,GPU}}^{(b)}$.

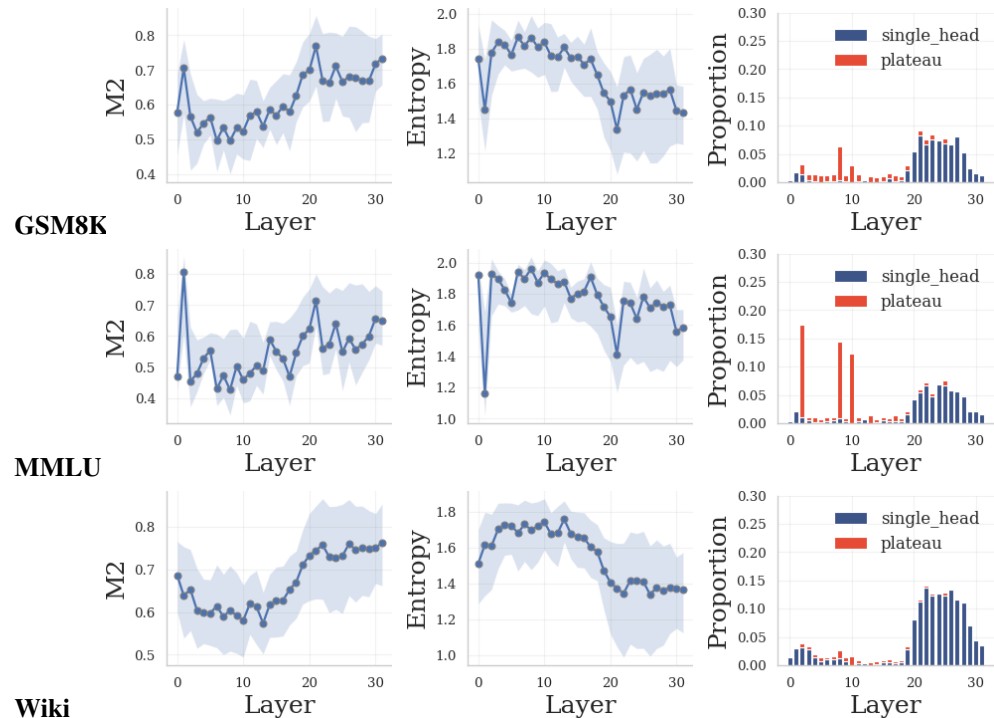

Figure 1: Layer-wise gate score distribution variability in Mixtral-8×7B. Rows correspond to datasets (GSM8K, MMLU, Wiki).

## 3.2 EXPERT-LOAD IMBALANCE AT INFERENCE

DeepSeek-MoE (Dai et al., 2024) adopts a loss-free balancing strategy during training by dynamically adjusting expert-specific biases in the gate scores to distribute tokens more evenly. Unlike auxiliary load-balancing losses, this approach avoids introducing gradient signals that may interfere with the primary objective. Inference traces, however, show imbalance. Evaluating DeepSeek-MoE-16b-chat on GSM8K, Figure 7a in Appendix A reports the per-layer token-weighted imbalance factor $I$. As shown, training-time balancing does not ensure balanced expert usage at inference.

## 3.3 LAYER VARIABILITY IN GATE SCORE DISTRIBUTIONS

To study the behavior of MoE routing, we analyze per-token gate score distributions across layers of Mixtral-8×7B ($k = 2$), where $k$ is the number of experts selected per token. We use three datasets: Wiki (Merity et al., 2016), GSM8K (Cobbe et al., 2021), and MMLU (Hendrycks et al., 2021), with 10,000 samples for prefill and 2,000 for decode.

Across three datasets, we summarize how routing behavior varies across layers by measuring three statistics: **Top-$k$ mass ($M_k$):** the fraction of probability assigned to the two most likely experts for each token, averaged across tokens. For Mixtral-8×7B, we compute $M_2$ since it has = 2. High $M_2$ indicates skewed routing with sharp dominance. **Entropy:** the Shannon entropy of the gate score distribution. Higher entropy means a flatter distribution where the probability is spread across multiple experts. **Routing regimes:** we classify distributions into three regimes: *single-head* (one expert strongly dominates), *plateau* (several experts have comparable scores near the top), and *smooth* (probability mass is more evenly distributed). These routing regimes map to natural actions. For single-head, we must assure that expert is chosen; for plateau, we have some flexibility in balancing load and the best expert choices for smooth, we have significant flexibility and can focus on load.

Across the three datasets, we quantify how routing behavior varies across layers using three statistics. **Top-$k$ mass ($M_k$):** the fraction of probability assigned to the $k$ most likely experts for each token, averaged across tokens. For Mixtral-8×7B we report $M_2$, since $k = 2$. A high $M_k$ indicates skewed routing with sharp dominance. **Entropy:** the Shannon entropy of the gate score distribution, where higher values reflect flatter distributions spread across more experts. **Routing regimes:**

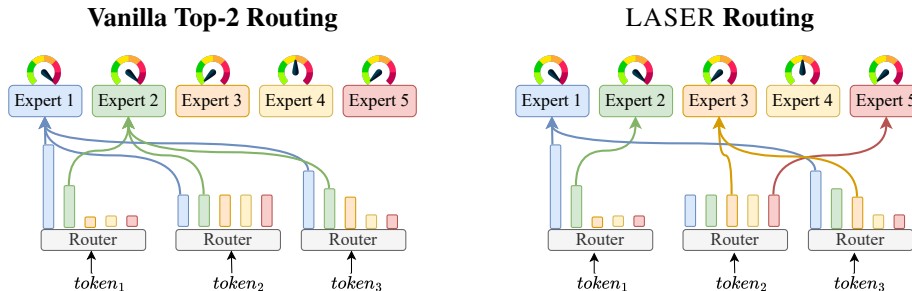

Figure 2: Comparison of routing strategies. Each figure shows 5 experts with $k = 2$ experts per token; the icon above each expert indicates load. Left (Vanilla): top-$k$ routing always picks the two highest-scoring experts (e.g., experts 1 and 2), even if they are overloaded. Right (LASER): routing adapts to score distribution and load. For token 1, the skewed distribution leads to the top-2 choice; for token 2, the uniform distribution lets LASER assign to the least-loaded experts; for token 3, expert 3's score is close to expert 2 and it is less loaded, so LASER selects experts 1 and 3.

we classify distributions into three regimes: *single-head* (one expert strongly dominates), *plateau* (several experts have comparable scores near the top), and *smooth* (probability mass is more evenly distributed). These regimes map to natural actions. For single-head, we must ensure that an expert is chosen; for plateau, we have some flexibility in balancing load, and the best expert choices for smooth operation allow for significant flexibility, enabling us to focus on load management.

Figure 1 reports results for Mixtral-8×7B on GSM8K, MMLU, and Wiki. Across all three datasets, we observe a consistent pattern: early and late layers exhibit skewed routing with high top-2 mass and low entropy, while middle layers are flatter with higher entropy and lower $M_2$, offering more flexibility for balancing. GSM8K shows the smoothest distributions, with broad token spread in the middle layers and clear opportunities for load-aware routing. MMLU displays sharper contrasts, with strongly skewed early and late layers. Wiki is the most skewed overall, concentrating heavily on a few experts at both the beginning and end of the network. These results indicate that balancing opportunities vary by layer and dataset, with middle layers generally offering the most headroom for routing-time optimization.

## 4 THE METHOD

We present LASER, a dynamic routing algorithm that balances expert load with answer quality. Existing methods always select a fixed top-$k$ experts per token, regardless of the gate score distribution. This inflexibility misses an important signal of the score distribution. When the score distribution is uniform, for example, all experts are equally relevant, so LASER can focus solely on load balancing without impacting accuracy. On the other hand, when the distribution is highly skewed, LASER limits the selection to the top-scoring experts, preserving the quality of the response. Figure 2 shows a comparison of vanilla MoE and LASER routing.

Building on our analysis in Section 3.3, we design LASER to adapt to distribution shape at each layer. When the top-$k$ experts already dominate the mass, LASER reverts to standard top-$k$ routing. Otherwise, it expands the candidate pool by including all experts above a threshold relative to the top score. To limit overhead, the pool is trimmed to the top-$c$ candidates, and the token is then assigned to the $k$ least-loaded experts. This procedure preserves relevance while actively balancing load. Algorithm 1 in Appendix B presents its pseudocode.

**Problem Formulation.** LASER routes each token by jointly preserving the gate score distribution and reducing expert load. Let $n$ denote the number of experts and $k$ the target number of active experts per token. Each expert $e \in \{1, \ldots, n\}$ has a current load $L_e$, and the router ultimately returns a set $A \subseteq \{1, \ldots, n\}$ with $|A| = k$.

For a token with gate score distribution $s \in \mathbb{R}^n$, we require $s_i \geq 0$ and $\sum_{i=1}^n s_i = 1$. Writing the scores in descending order, $s_{(1)} \geq s_{(2)} \geq \cdots \geq s_{(n)}$, we define the *top-k mass* $M_k = \sum_{i=1}^k s_{(i)}$. This statistic summarizes distribution shape: high $M_k$ indicates sharp dominance, while low $M_k$ indicates flatter distributions.

**Overview of** LASER. Given gate scores and current expert loads, LASER routes each token in three stages. First, it decides whether to expand beyond the static top-$k$ experts based on the score distribution. Second, if expansion occurs, it constructs a candidate pool and may trim it to reduce selection cost. Finally, it assigns the token to the $k$ least-loaded experts within the working set.

**Expansion rule.** If the top-$k$ scores already dominate, expansion is unnecessary. Given a high-mass threshold $\varepsilon_{\text{high}} \in (0, 1)$, if $M_k \geq \varepsilon_{\text{high}}$, LASER routes directly to the top-$k$ experts and stops. This preserves accuracy in skewed cases and avoids unnecessary computation. Otherwise, LASER expands beyond the static top-$k$. It sets a cutoff tied to the maximum score, $t = t_{\text{fix}} \cdot s_{(1)}$, with $t_{\text{fix}} \in (0, 1]$, and forms the candidate pool $\mathcal{T} = \{ i \in [n] : s_i \geq t \}$. By construction, $m = |\mathcal{T}| \geq k$, and the top-$k$ experts are always included. This ensures the standard baseline is preserved while admitting only experts whose scores are within a fixed fraction of the leader $s_{(1)}$.

**Final assignment.** From the candidate pool $\mathcal{T}$ of size $m$, LASER optionally trims to a working set of size $c$ with $k \leq c \leq m$. Trimming can be deterministic (keeping the $c$ highest-scoring experts) or randomized (sampling $c$ uniformly). It then assigns the token to the $k$ least-loaded experts based on $\{L_e\}$, breaking ties by score. Setting $c = k$ recovers standard top-$k$ routing, while larger $c$ values allow more flexibility in balancing at the cost of slightly higher selection work ($O(c \log c)$ per token).

**Parameter setting and layer-wise choices.** LASER has three knobs: the dominance cutoff $\varepsilon_{\text{high}} \in (0, 1)$, the pool threshold $t_{\text{fix}} \in (0, 1]$, and the trimming size $c \in \{k, \ldots, m\}$. The algorithm behaves conservatively when the distribution is skewed (large $M_k$) and expands when it is flat (small $M_k$). Figures 8 in Appendix B.1 compare prefill and decode statistics for entropy and $M_2$. The two phases follow similar trends across layers, with aligned medians and variability. This consistency allows parameters to be calibrated from prefill traces alone and then reused at decode. Prefill curves highlight which layers are typically skewed and which are flatter. When $M_k$ is high, a larger $\varepsilon_{\text{high}}$ discourages expansion, preserving accuracy. When entropy is higher or $M_k$ is lower, a smaller $\varepsilon_{\text{high}}$ encourages expansion. The parameter $t_{\text{fix}}$ then controls the width of the candidate pool: higher values keep the pool closer to the top experts, while lower values admit more candidates. Trimming size $c$ bounds the selection cost while allowing some flexibility.

LASER also supports layer-specific settings. At one extreme, a single global configuration can be used across all layers; at the other, each layer can have its own parameters. Our analysis (Section 3.3) shows that early, middle, and final layers display different gate score patterns. A practical compromise is to group layers into these three bands and assign band-specific values of $\varepsilon_{\text{high}}$ and $t_{\text{fix}}$.

**Integration with MoE Forward Pass.** We apply LASER immediately after the gate function in the MoE forward pass. Once gate scores are computed, LASER constructs the candidate pool and routes tokens based on both score distribution and expert load. This design allows plug-and-play integration without altering the base model parameters or training process.

**Beyond Load Balancing.** While LASER currently ranks candidates by load, the ranking criterion is modular. In principle, one could incorporate other system-aware objectives, such as locality (e.g., preferring experts on the same node), bandwidth constraints, or memory pressure. Exploring these objectives is left for future work, but the design highlights that LASER is not limited to reducing load imbalance and can serve as a general plug-and-play framework for routing-time optimization.

## 5 EVALUATION

**Models and Datasets.** We evaluate LASER on two MoE language models: Mixtral-8×7B with $k = 2$ active experts per token, and DeepSeek-MoE-16b-chat with $k = 6$. (We use the same values of $k$ as in the original training of these models.) We evaluate on four datasets spanning different domains: (i) MMLU (57 subjects, multiple-choice; 512 samples), (ii) GSM8K (grade-school mathematics word problems; 256 samples), and (iii–iv) ARC-Easy and ARC-Challenge (science multiple-choice questions; 512 samples each, with ARC-Challenge representing the harder subset). Together, these datasets cover reasoning, factual knowledge, and science problem solving, offering a broad testbed for inference-time routing. The evaluation was conducted on a server equipped with one NVIDIA A100 80GB PCIe GPUs, one AMD EPYC 7313 16-Core processors, and 504 GiB of RAM. Experiments used PyTorch 2.6.0 together with Transformers 4.48.3.

**Integration.** To integrate our custom load-balancing router, we identify all MoE layers with gating modules and override their forward methods. The new forward pass invokes our gating function,

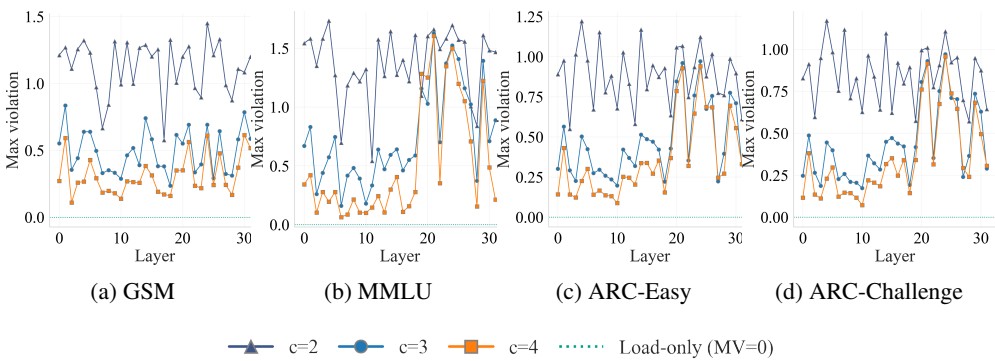

(a) GSM  (b) MMLU  (c) ARC-Easy  (d) ARC-Challenge

Figure 3: Per-layer max violation (MV) on Mixtral-8×7B ($k = 2$) across GSM8K, MMLU, ARC-Easy, and ARC-Challenge. We compare LASER with candidate pool sizes $c = 2, 3, 4$ against the load-only lower bound (MV=0). Increasing $c$ reduces MV across most layers, with the largest improvements in middle layers where gate score distributions are flatter. In contrast, the final layers have high top-$M_2$ mass; our setting of $\varepsilon_{\text{high}}$ disables expansion in these layers, so MV remains unchanged but accuracy is preserved.

which computes router logits and assigns experts to tokens using our policy. The remaining pipeline, including attention and feed-forward computation, remains unchanged. This design preserves the architecture and weights while transparently substituting our gating decision at inference time. $\varepsilon_{\text{high}}$ and $t_{\text{fix}}$, for each model and dataset, are set as described in Table 1 (Appendix B.2).

**Metrics.** To capture load imbalance, we report expert imbalance rather than GPU imbalance. This choice reflects our evaluation setup, which used small models on a single GPU due to resource limits, making expert imbalance the natural metric to track. For deployment relevance, Section 3.1 introduces a mapping from expert imbalance to GPU imbalance that accounts for expert placement and replication. Reporting expert imbalance allows for the translation of GPU imbalance under any deployment. We use two related metrics. The first is the *imbalance factor* ($I_{\text{agg}}$), which measures how unevenly tokens are distributed across experts, aggregated over layers. A higher value indicates that some experts are overloaded relative to others. We summarize $I_{\text{agg}}$ using the 50th percentile (P50, median across batches) and the 95th percentile (P95, tail imbalance across batches). The second is the *max violation* (MV), which quantifies how much the most loaded expert in a layer exceeds the average. Lower values of both metrics correspond to more balanced expert utilization and, consequently, lower inference latency and cost. In addition, we report score (accuracy) as the fraction of correctly answered questions.

**Baselines.** We compare against two routing strategies: (i) **Vanilla Top-$k$**: the default MoE policy that selects the $k$ experts with highest gate scores. (ii) **Load-only**: ignores gate scores and routes tokens solely to balance expert load. This achieves near-perfect load balancing at the cost of accuracy.

**Per-layer imbalance.** Figure 3 reports per-layer max violation (MV) on Mixtral-8×7B. We compare LASER with candidate pool sizes $c \in \{2, 3, 4\}$ to vanilla top-$k$ ($c = k = 2$) and the load-only lower bound (MV= 0). Guided by the layerwise score-distribution analysis (Section 3.3), we set $\varepsilon_{\text{high}}$ and $t_{\text{fix}}$ separately for early, middle, and final layers. Increasing $c$ reduces MV across datasets, with the largest improvements in the middle layers where score distributions are flatter. In the final layers, where gate distributions are highly skewed and the top-$M_2$ mass is large, our setting of $\varepsilon_{\text{high}}$ disables expansion. As a result, MV remains unchanged, but accuracy is unaffected (see Figs. 5–6). Overall, most of the balancing gains come from the middle layers. By contrast, vanilla top-$k$ ($c = 2$) shows consistently higher MV with sharp fluctuations across depth, indicating persistent imbalance. Figure 3 reports per-layer max violation (MV) on Mixtral-8×7B; DeepSeek results appear in Appendix B.3. We compare LASER with candidate pool sizes $c \in \{2, 3, 4\}$ to vanilla top-$k$ ($c = k = 2$) and the load-only lower bound (MV= 0). Guided by the layerwise score-distribution analysis (Section 3.3), we set $\varepsilon_{\text{high}}$ and $t_{\text{fix}}$ separately for early, middle, and final layers. Increasing $c$ reduces MV across datasets, with the largest improvements in the middle layers where score distributions are flatter. In the final layers, where gate distributions are highly skewed and the top-$M_2$ mass is large, our setting of $\varepsilon_{\text{high}}$ disables expansion. As a result, MV remains unchanged, but

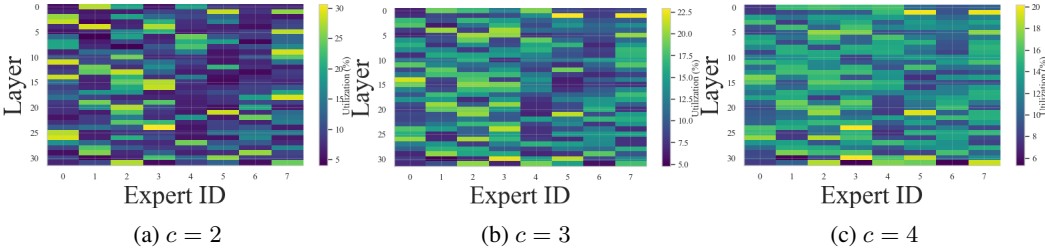

(a) $c = 2$      (b) $c = 3$      (c) $c = 4$

Figure 4: Expert utilization in Mixtral-8×7B on GSM8K for different candidate pool sizes ($c = 2, 3, 4$). For $c = 2$, token assignments concentrate on a few experts, leading to imbalance. As $c$ increases, tokens spread more evenly across experts, producing smoother utilization patterns.

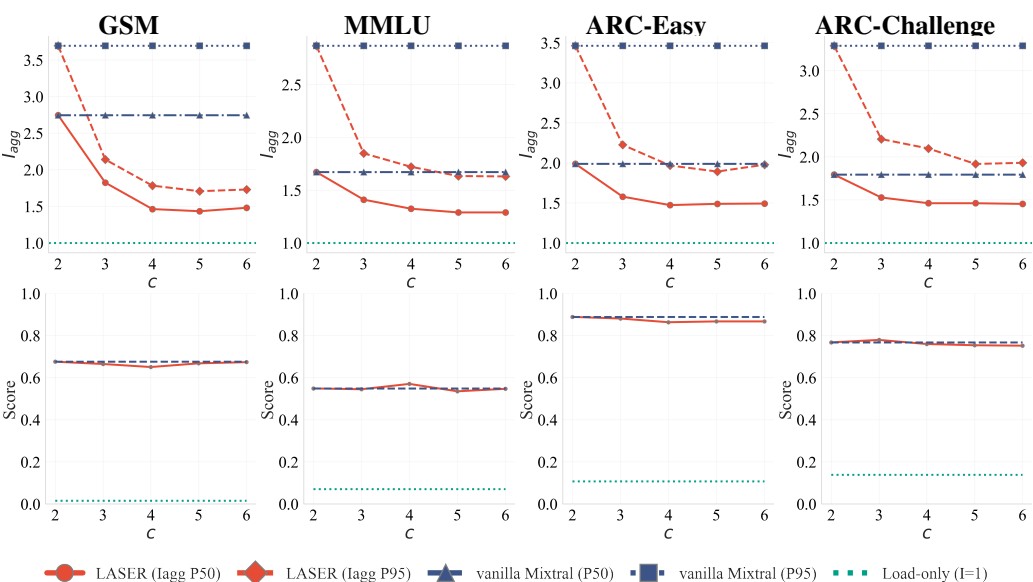

Figure 5: Mixtral-8×7B ($k = 2$). LASER maintains accuracy while reducing imbalance ($I_{\text{agg}}$) across datasets. The largest improvement appears on GSM8K (up to $1.63\times$ reduction in mean $I_{\text{agg}}$). When $c = k$, LASER matches vanilla top-$k$ routing.

accuracy is unaffected (see Figs. 5–6). Overall, most of the balancing gains come from the middle layers. By contrast, vanilla top-$k$ ($c = 2$) shows consistently higher MV with sharp fluctuations across depth, indicating persistent imbalance.

Figure 4 visualizes expert utilization in Mixtral-8×7B on GSM8K for different candidate pool sizes ($c = 2, 3, 4$). Results for the other datasets and for the DeepSeek model across all four datasets are provided in Appendix B.4. Expert utilization refers to the number of tokens assigned to each expert in each layer. Each heatmap plots layers on the vertical axis and expert IDs on the horizontal axis, with color intensity indicating the relative load of a given expert in a given layer. When $c = 2$, token assignments are concentrated on a few experts, producing strong color contrasts that reflect imbalance. As $c$ increases, assignments spread more evenly across experts, leading to smoother heatmaps with reduced concentration of load. Final layers remain skewed because their gate score distributions have large top-$M_2$ mass, and our parameter setting disables expansion in those layers.

**Score and Load.** We integrate LASER into both Mixtral-8×7B ($k = 2$) and DeepSeek-MoE-16b-chat ($k = 6$), and evaluate performance across four datasets. Figures 5 and 6 report accuracy (Score) and imbalance ($I_{\text{agg}}$) as we vary the candidate pool size $c$. When $c = k$, LASER reduces to standard top-$k$ routing, and the curves for LASER and vanilla coincide. Increasing $c$ improves load balance by distributing tokens across a larger set of experts.

Across all datasets, LASER maintains accuracy comparable to vanilla routing, with absolute differences below $0.02$, while consistently reducing imbalance. We summarize $I_{\text{agg}}$ using the 50th

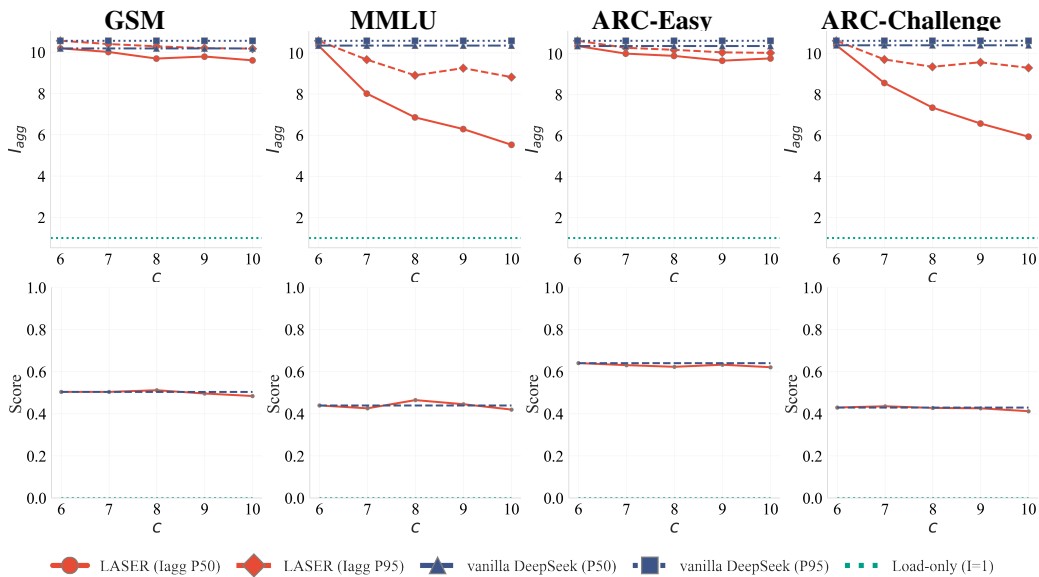

Figure 6: DeepSeek-MoE-16B-Chat ($k = 6$). LASER preserves accuracy while reducing imbalance ($I_{\text{agg}}$). The largest improvements occur on ARC-Challenge and MMLU, each with about $1.4\times$ reduction in mean $I_{\text{agg}}$. As in Mixtral, LASER converges to vanilla top-$k$ routing when $c = k$.

percentile (P50, median across batches) and 95th percentile (P95, tail imbalance across batches). On Mixtral, the largest reduction occurs on GSM8K, with a $1.63\times$ improvement in mean $I_{\text{agg}}$. On DeepSeek, the largest improvements appear on ARC-Challenge and MMLU, with reductions of about $1.4\times$. The load-only baseline achieves perfect balance ($I_{\text{agg}} = 1$) but fails entirely on accuracy, confirming the need to preserve score quality. Because decoding step time is governed by the most heavily loaded expert or GPU (Section 3.1), these reductions in imbalance directly translate into lower average latency and improved throughput in deployment.

LASER provides benefits even when the total number of experts is small, as in Mixtral (8 experts, $k = 2$ active per token). Even modest expansions of the candidate pool ($c > k$) reduce $I_{\text{agg}}$ while leaving accuracy unchanged. In DeepSeek, with 64 total experts and $k = 6$ active per token, LASER similarly reduces imbalance. Importantly, $c$ acts only as an upper bound on the candidate pool size. The actual number of candidates for a token is determined by the thresholding rule controlled by $\varepsilon_{\text{high}}$ and $t_{\text{fix}}$. If the gate distribution is highly skewed, few experts exceed the threshold, and the pool size remains small even for large $c$. This explains why accuracy does not decrease and why load-balance gains saturate as $c$ increases, as seen in Figure 5 for $c = 5, 6$. Careful calibration of $\varepsilon_{\text{high}}$ and $t_{\text{fix}}$ is therefore essential.

# 6 CONCLUSION AND LIMITATIONS

We introduced LASER, a plug-and-play inference-time routing algorithm for Mixture-of-Experts models that balances load while preserving accuracy. LASER adapts to the shape of the gate score distribution: when scores show a clear preference, it routes to the strongest experts; when scores are more uniform, it broadens the candidate set and selects the least-loaded experts. Our evaluation shows that LASER consistently reduces expert-load imbalance across Mixtral and DeepSeek models and four datasets. Because LASER operates directly on gate scores, it integrates seamlessly into existing inference pipelines without retraining, and its design extends naturally to other system-level objectives beyond load balancing.

**Limitations and Future Directions.** One limitation of this work is parameter setting. In our evaluation, we set $\varepsilon_{\text{high}}$ and $t_{\text{fix}}$ per dataset using prefill only. Future directions include developing methods to predict parameters automatically from dataset characteristics or to let the system warm up and adjust them dynamically at runtime. Another potential approach is to extend the framework to account for communication locality, memory pressure, and other system-level constraints.

# 7    ADDITIONAL INFORMATION

**Ethics statement.** Our study does not raise any ethical concerns.

**Reproducibility statement.** The evaluation datasets (GSM8K, MMLU, ARC-Easy, and ARC-Challenge) and the LLMs (Mixtral-8×7B and DeepSeek-MoE-16b-chat) are publicly available on HuggingFace. The codes to replicate the evaluations are provided.

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

# Part I

# Appendix

## A EXPERT-LOAD IMBALANCE AT INFERENCE

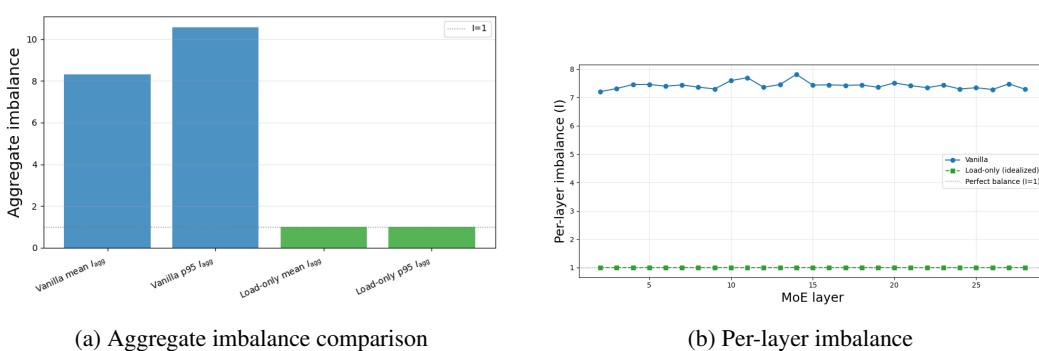

(a) Aggregate imbalance comparison

(b) Per-layer imbalance

Figure 7: per-layer token-weighted imbalance factor $I$

Figure 7a shows the per-layer token-weighted imbalance factor $I$. Each point corresponds to one MoE layer, compared to routing by load only (with $I = 1$), which represents a perfect balance. Vanilla DeepSeek-MoE-16b-chat routing (blue) yields imbalance values around 77–88 across layers, far from the idealized "load-only" baseline (green), which achieves near-perfect balance by ignoring scores. Figure 7b summarizes aggregate imbalances across layers and batches. The vanilla model shows high mean and tail imbalances ($I_{agg}$ p95) exceeding 10, confirming that large deviations persist during inference. In contrast, the load-only baseline maintains values close to 1, demonstrating how much headroom remains if routing could better account for load.

## B LASER PSEUDO CODE

---

**Algorithm 1** LASER (per token, per layer)

---

**Require:** scores $s \in \mathbb{R}^n$ with $s_i \geq 0$ and $\sum_i s_i = 1$; loads $L \in \mathbb{N}^n$;
   parameters: $k$ (targets), $\varepsilon_{\text{high}} \in (0,1)$, $t_{\text{fix}} \in (0,1]$, $c \in \{k, \ldots, n\}$;
   trimming mode MODE $\in \{\text{TOP}, \text{RANDOM}\}$.
1: compute top-$k$ mass: $M_k \leftarrow \sum_{i=1}^{k} s_{(i)}$
2: **if** $M_k \geq \varepsilon_{\text{high}}$ **then**               ▷ distribution is sharply skewed
3:   $A \leftarrow \{(1), \ldots, (k)\}$             ▷ use baseline top-$k$ only
4:   **return** $A$
5: **end if**
6: $t \leftarrow t_{\text{fix}} \cdot s_{(1)}$              ▷ fixed cutoff tied to the max score
7: $\mathcal{T} \leftarrow \{i \in [n] : s_i \geq t\} \cup \{(1), \ldots, (k)\}$       ▷ ensure $|\mathcal{T}| \geq k$
8: $m \leftarrow |\mathcal{T}|$
9: $c^\star \leftarrow \min(c, m)$
10: **if** MODE $=$ TOP **then**
11:   Cand $\leftarrow$ the $c^\star$ indices in $\mathcal{T}$ with largest $s_i$
12: **else**                    ▷ MODE $=$ RANDOM
13:   Cand $\leftarrow$ uniformly sample $c^\star$ indices from $\mathcal{T}$ (without replacement)
14: **end if**
15: Cand $\leftarrow$ STABLESORTBYLOAD(Cand; $L$)  ▷ primary key: ascending $L_e$; tiebreak: descending $s_e$
16: $A \leftarrow$ first $k$ indices of Cand
17: **return** $A$

---

## B.1 PREFILL AND DECODE STATISTICS FOR ENTROPY AND M2.

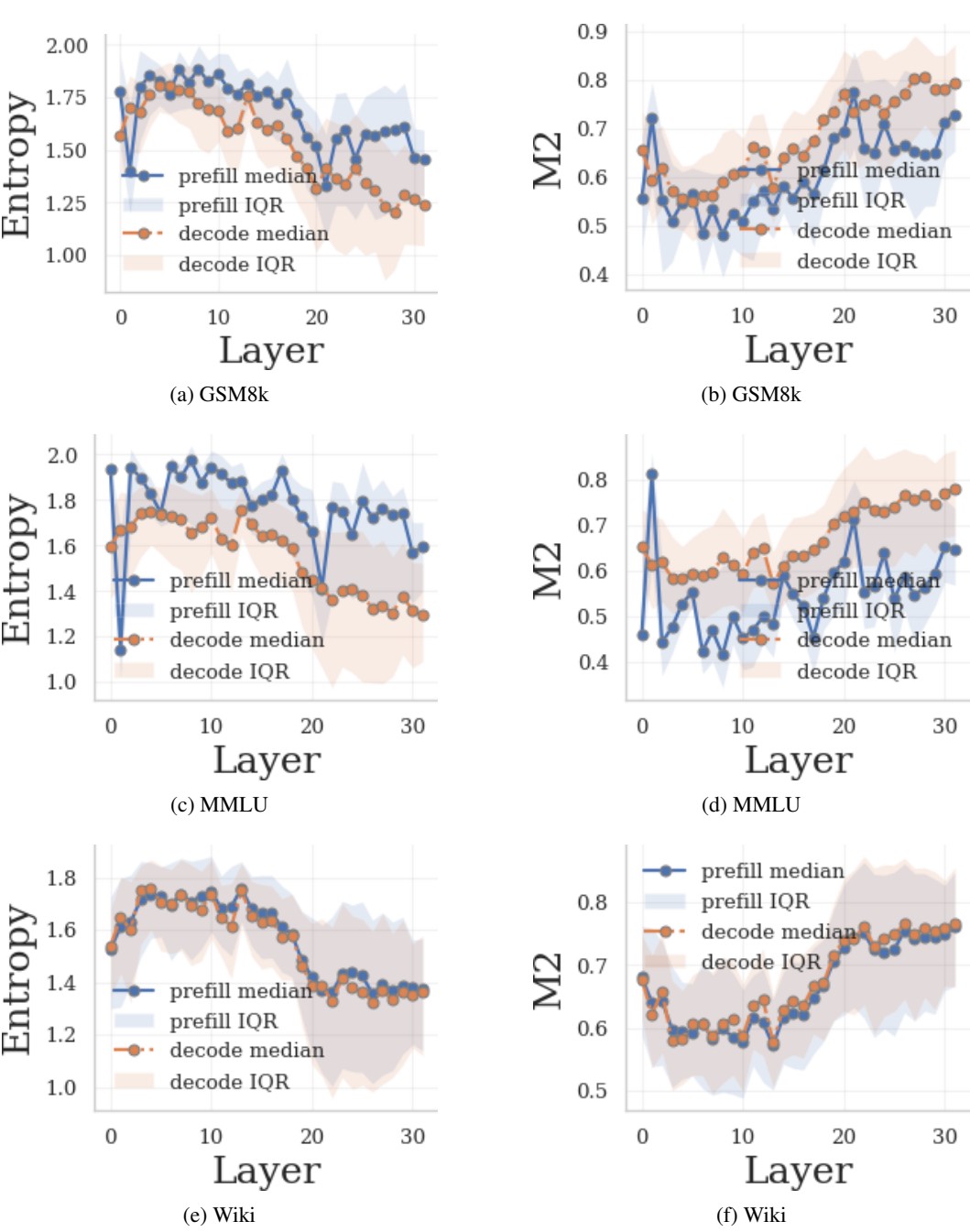

(a) GSM8k      (b) GSM8k

(c) MMLU      (d) MMLU

(e) Wiki      (f) Wiki

Figure 8: Prefill and decode statistics for entropy and M2 using Mixtral-8×7B across GSM8k, MMLU and Wiki datasets

## B.2 USED PARAMETERS FOR EVALUATION

.

We use the following parameter settings for each model and dataset, derived from our analysis of prefill statistics ($M_2$ and entropy). As discussed in Section 3.3, early, middle, and final layers exhibit distinct gate score patterns, which motivates band-specific thresholds for $\varepsilon_{\text{high}}$ and $t_{\text{fix}}$.

| Model | Dataset | $t_{\text{fix}}$ (E/M/F) | $\varepsilon_{\text{high}}$ (E/M/F) |
|---|---|---|---|
| DeepSeek–MoE–16B–Chat | ARC Challenge | 0.80/0.80/0.80 | 0.40/0.40/0.40 |
| DeepSeek–MoE–16B–Chat | ARC Easy | 0.80/0.80/0.80 | 0.35/0.35/0.35 |
| Mixtral–8×7B | ARC Challenge | 0.60/0.60/0.60 | 0.7159/0.6419/0.6285 |
| Mixtral–8×7B | ARC Easy | 0.60/0.60/0.60 | 0.7159/0.6419/0.6285 |
| DeepSeek–MoE–16B–Chat | GSM8K | 0.25/0.45/0.55 | 0.30/0.30/0.30 |
| DeepSeek–MoE–16B–Chat | MMLU | 0.80/0.80/0.80 | 0.40/0.40/0.40 |
| Mixtral–8×7B | GSM8K | 0.60/0.60/0.60 | 0.72/0.75/0.80 |
| Mixtral–8×7B | MMLU | 0.40/0.40/0.40 | 0.7159/0.6419/0.6285 |

Table 1: Thresholds used per model and dataset. Values shown as Early/Middle/Final layers.

### B.3 PER-LAYER MAX VIOLATION ON DEEPSEEK

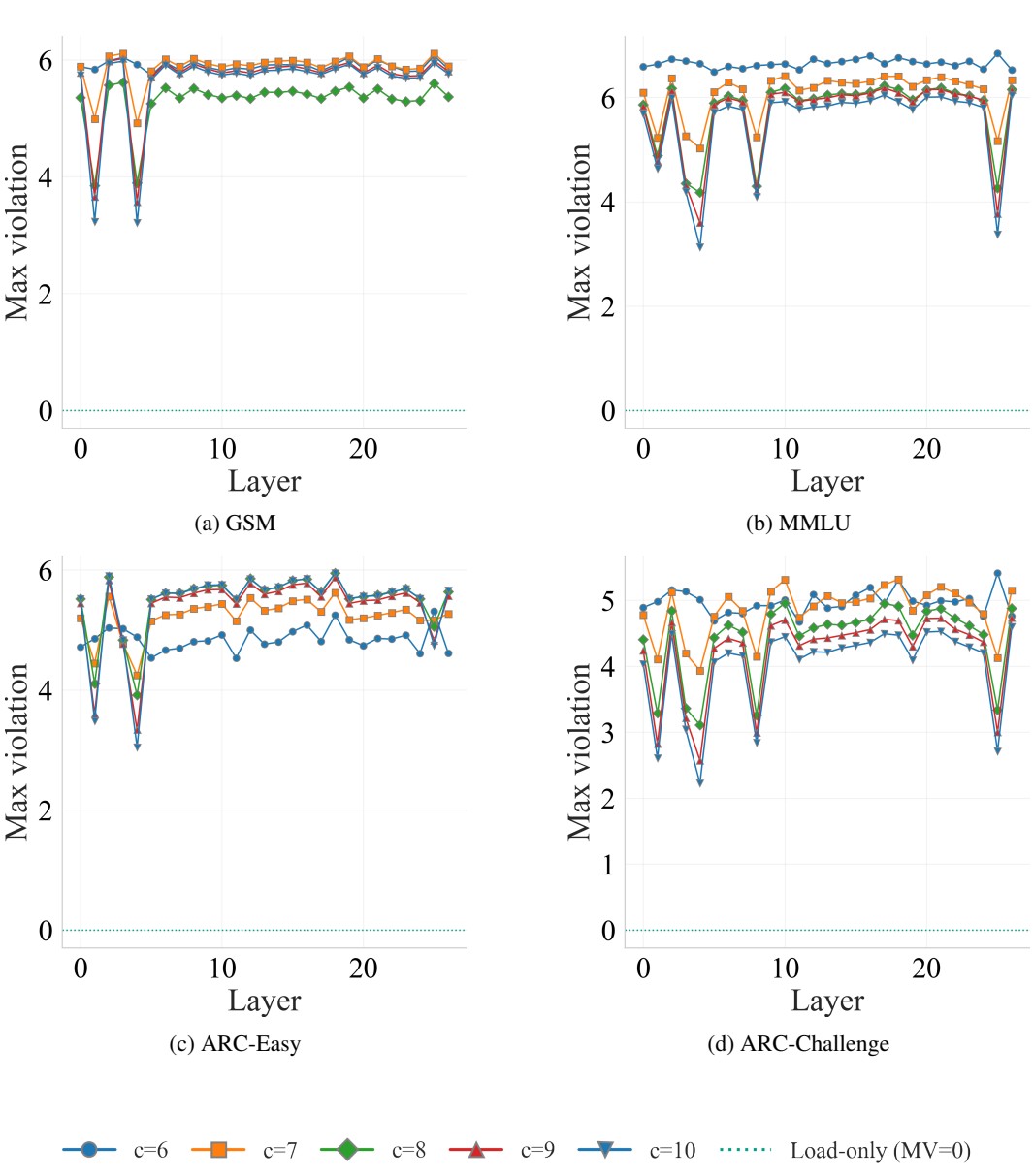

Figure 9: Per-layer max violation (MV) on DeepSeek-MoE across GSM8K, MMLU, ARC-Easy, and ARC-Challenge.

### B.4 VISUALIZATION OF EXPERT UTILIZATION USING MIXTRAL AND DEEPSEEK

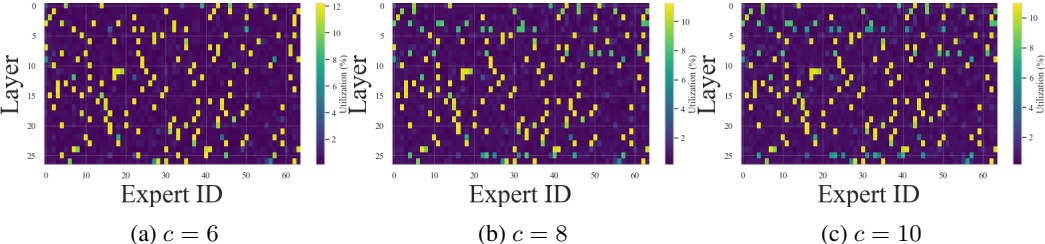

(a) $c = 6$  (b) $c = 8$  (c) $c = 10$

Figure 10: Expert utilization in DeepSeek on MMLU for different candidate pool sizes ($c = 6, 8, 10$).

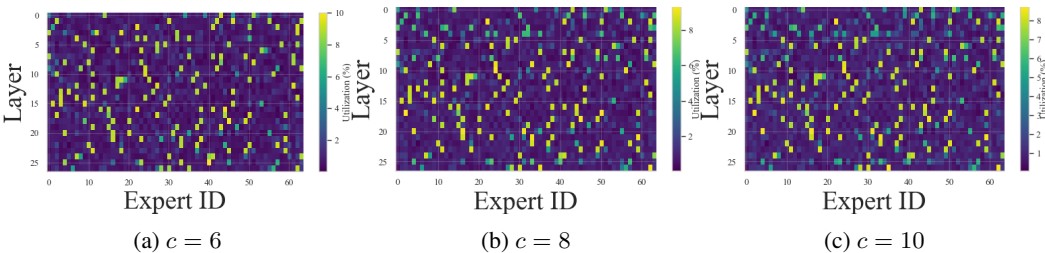

(a) $c = 6$  (b) $c = 8$  (c) $c = 10$

Figure 11: Expert utilization in DeepSeek on ARC-Challenge for different candidate pool sizes ($c = 6, 8, 10$).

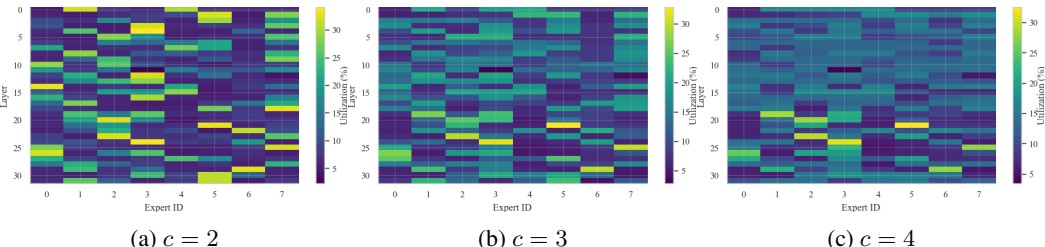

(a) $c = 2$  (b) $c = 3$  (c) $c = 4$

Figure 12: Expert utilization in Mixtral-8×7B on MMLU for different candidate pool sizes ($c = 2, 3, 4$).

### B.5 THE USE OF LARGE LANGUAGE MODELS (LLMS)

We use LLMs to condense writings and fix grammatical errors.

