# OpenReview forum: "From Score Distributions to Balance:  Plug-and-Play Mixture-of-Experts Routing"
_ICLR.cc/2026/Conference — ICLR 2026 Conference Withdrawn Submission_

### Official Review · Reviewer_FZA3 · 2025-10-31

**Soundness:** 2
**Presentation:** 1
**Contribution:** 2
**Rating:** 2
**Confidence:** 3

**Summary:**

The paper proposes plug-and-play routing algorithm for MoEs at inference time. It aims to solve the expert load imbalance (where some GPUs are overloaded, others are idle) through the gate score distribution for each token. If the scores are peaked (favoring a few experts), it uses the standard top-k experts. If the scores are flat (spread out), it creates a candidate pool of relevant experts. From this pool, it routes the token to the k least-loaded experts to balance the model.

**Strengths:**

This paper proposes LASER, an inference-time routing algorithm for reducing expert load imbalance in MoE models. While the problem of load imbalance is indeed important, the contributions and presentation of the proposed model are weak.

**Weaknesses:**

Although the paper addresses a relevant problem, the way it presents the motivation, and the disconnect between its goals and the actual algorithmic behavior, makes the proposed model difficult to understand. Please see the question block.

**Questions:**

The motivation is based on two unproven assumptions.  First, the paper claims that in the case of {\it{smooth}} distributions, we have significant flexibility in expert selection and can focus on load. Indeed, this assumes that the small differences in expert scores are meaningless, but there is no evidence (some proofs/propositions, theories, related references) to support this. Based on existing methods, the output of a gating network is a learned and meaningful distribution, so even minor/small differences can be significant. It is technically unsound to say that a smooth distribution (e.g., Expert A: 30%, Expert B: 28%, Expert C: 27%) is a signal from the model that all these experts are equivalent.  A delta of 1-3% may be a distinction between a correct and incorrect token generation. There is some misunderstanding in this motivation, so needs justification.

Second, the paper bases its latency model on $T_{step} \sim \gamma * I_{agg,GPU} + T_{comm} + T_{offload}$, and then states that  latency scales nearly linearly with $I_{agg,GPU}$ and that throughput improvements are tied to reducing this imbalance. But, this assumption, \gamma *I_{agg,GPU} (the GPU aggregation index), may not valid for large MoEs. I think the main bottlenecks are the other terms; e.g.,  T_{comm}, which represents the cost of shuffling tokens across GPUs.  Based on this claim, one can argue that even if the model balances the expert load (reducing I_{agg,GPU} to 1.0), the latency will see almost no improvement because the $T_{comm}$ bottleneck will still exist. Even the paper acknowledges that the model only holds when $C = T_{comm} + T_{offload}$ is small, which is the weakness, because in large transformers and datasets C is almost not small.  I suggest the authors to provide stronger justification for this modeling assumption or revise the motivation.

In Section 4, the authors state that the algorithm routes each token by jointly preserving the gate score distribution and reducing expert load. Then, in the final assignment stage, the model assigns the token to the k least-loaded experts based on $L_e$, ignoring the score. But, this is not a joint optimization. The algorithm uses scores to create a candidate pool, then completely ignores the scores when making the final decision. This means that tokens can be routed to the lowest-scoring experts in the candidate set, as long as they are underutilized. So, this is opposite of the paper’s main goal of maintaining gate score fidelity and accuracy.

---

### Official Review · Reviewer_VPDF · 2025-10-31

**Soundness:** 2
**Presentation:** 3
**Contribution:** 2
**Rating:** 2
**Confidence:** 4

**Summary:**

In this paper the authors introduce LASER, a plug-and-play, inference-time routing algorithm for MoE LLMs that adapts the expert candidate set to the gate score distribution. The method then assigns tokens to the least-loaded experts to reduce load imbalance while preserving accuracy. The main claim is that when the top-k mass indicates a sharply peaked distribution, it means the router is highly confident and hence LASER should fall back to vanilla top-k, otherwise it thresholds scores, trims to generate top-c candidates, and routes to the least-loaded k experts. The authors characterize gate score distribution variability across layers and datasets, motivating adaptive expansion mainly in middle layers. The method is validated for Mixtral and DeepSeek‑MoE‑16B models across ARC‑Easy/Challenge, MMLU, and GSM8K. The proposed method reduces expert-imbalance metrics with negligible accuracy change, and argues these improvements translate to lower latency in deployment.

**Strengths:**

- The method is simple, clear to understand, and easy to integrate into existing MoE forward passes without retraining.
- Reducing imbalance at inference without model retraining is valuable for MoE LLM serving.
- The layer-wise analysis of gate score distributions identifies middle layers as the main headroom for balancing which is intuitive.

**Weaknesses:**

**Insufficient Baselines and Limited Novelty:** The primary weakness of this work is the lack of comparison to relevant and more sophisticated baseline methods. The evaluation only includes "vanilla top-k" and an accuracy-agnostic "load-only" router which only serve to establish lower and upper bounds on performance. This is insufficient to demonstrate the method's effectiveness. The authors should include experimental comparisons against other dynamic routing or load-balancing techniques discussed in the related work, such as *Ada-K* [1] , or other recent methods that define candidate sets of experts. For example, the techniques for defining top-c candidate sets in recent work like "Mixture of cache conditional experts" [2] would serve as highly relevant and strong baselines (see the 3 methods proposed which all could be relevant). Without these comparisons, the novelty and performance of the proposed simple heuristic are not well-contextualized.

**Lack of Direct Performance Metrics**: The evaluation solely relies on "expert imbalance" as a proxy metric for performance but fails to provide direct evidence of improved inference speed. The assumption of a linear relationship between imbalance and latency is not empirically validated in the paper. Furthermore, the computational overhead of the LASER algorithm itself is not measured. The authors should provide end-to-end performance metrics on their hardware setup. This should include wall-clock latency and throughput measurements to prove that the reduction in imbalance translates to a tangible speedup in a real-world deployment scenario. This is critical for substantiating the paper's central claim as this is mainly an empirical study.

**Parameter Sensitivity and Practicality Concerns**: The method introduces hyperparameters ($\epsilon_{high}$ and $t_{fix}$) that should be tuned for each model and dataset, as shown in Appendix B.2. This contradicts the "plug-and-play" nature of the algorithm and introduces a practical hurdle for adoption. While developing a fully automated tuning method may be beyond the scope of this work, the authors should provide a sensitivity analysis for these parameters.

**Limited Scope of Experiments**: Given that this study is primarily empirical and the method is based on heuristics, the paper needs a stronger empirical study. The evaluation is conducted on only two MoE models. While these are relevant, the field is evolving quickly. The authors should aim to include at least one more recent or architecturally distinct MoE model in their evaluation such as Qwen3-30B-A3B.


Finally, while the paper presents an interesting idea for a practical problem, the limited novelty, insufficient experimental validation, and lack of strong baselines make it fall short of the standards for ICLR. The work is better suited for a systems-focused conference, provided the authors address the critical need for wall-clock performance benchmarks.


[1] Yue, Tongtian, et al. "Ada-k routing: Boosting the efficiency of moe-based llms." The Thirteenth International Conference on Learning Representations. 2024.

[2] Skliar, Andrii, et al. "Mixture of Cache-Conditional Experts for Efficient Mobile Device Inference." Transactions on Machine Learning Research. 2025.

**Questions:**

See the weaknesses.

---

### Official Review · Reviewer_7Hvj · 2025-10-31

**Soundness:** 2
**Presentation:** 3
**Contribution:** 2
**Rating:** 4
**Confidence:** 5

**Summary:**

This paper introduces LASER, a plug-and-play routing algorithm for Mixture-of-Experts (MoE) models that operates at inference time. LASER dynamically adjusts the candidate pool of experts based on the shape of the gate score distribution—narrowing it when scores are skewed and expanding it when scores are uniform—then selects the final k experts based on real-time load. It requires no retraining or fine-tuning and can be directly integrated into existing models. Experiments on Mixtral-8×7B and DeepSeek-MoE-16B show that LASER reduces expert load imbalance by 1.4–1.9× while maintaining accuracy, leading to improved inference latency and throughput.

**Strengths:**

1. Addresses a real and pressing problem: inference-time load imbalance in MoE models, which directly hurts latency, throughput, and cost.
2. Plug-and-play: zero retraining or fine-tuning; needs only gate scores, so it drops into existing pipelines with a few lines of code.
Distribution-aware: adapts the candidate pool to the actual shape of gate scores (skewed vs. flat), preserving accuracy while gaining balance.
3. Expert-level granularity: balances at individual-expert scope, finer than prior GPU/node-level heuristics.
4. Strong empirical results: up to 1.9× reduction in load imbalance on two large models and four datasets with <0.02 accuracy drop.

**Weaknesses:**

1. Limited novelty: the core design of LASER is broadening the expert candidate pool and considering different tokens for a better-balanced MoE load. Similar ideas have been investigated in previous works, such as Occult (ICML 2025). You may clarify the differences between these papers.
2. Limited experiments: the experiments are only conducted on a single A100 GPU, while expert parallelism should be considered as a generalized and practical setting to validate LASER.
3. Lack of theoretical guarantees. LASER’s load-balancing improvement comes with no lower-bound or convergence analysis; only empirical evidence is offered. One could build a minimal model (e.g., Poisson arrivals + greedy routing) to derive the expected load gap between LASER and top-k, giving theoretical intuition.

**Questions:**

A significant drawback of this paper is the increase in communication cost. The gigantic size of the MoE layer makes it necessary to distribute it across different nodes in expert parallelism for very large-scale models such as DeepSeek V3 and Kimi K2. However, different nodes are typically interconnected with IB and possess relatively low bandwidth (~40GB/s). Therefore, in a real LLM serving cluster, the communication efficiency can be of significant importance, similar to the load-balancing problem. However, there is a native paradox in your paper, i.e, the load balancing in your paper is achieved at the cost of significant communication overhead. You may refer to Occult (ICML 2025) and consider how to better tame this native contradiction.

---

### Official Review · Reviewer_tApk · 2025-11-01

**Soundness:** 2
**Presentation:** 2
**Contribution:** 3
**Rating:** 4
**Confidence:** 3

**Summary:**

This paper introduces LASER, a plug-and-play, inference-time routing algorithm for Mixture-of-Experts (MoE) models. The goal is to address the critical problem of expert load imbalance during inference.

**Strengths:**

- Load imbalance is a primary bottleneck for inference, and the paper correctly identifies that system performance is gated by stragglers.

- No Retraining ("Plug-and-Play"). By operating purely at inference time using only the trained model's gate scores, proposed LASER can be applied to existing, pre-trained MoE models.

- The evaluation shows a clear and consistent reduction in load imbalance metrics.

**Weaknesses:**

- Lack of Direct System Performance Evaluation. This paper claims about improving "latency, throughput, and cost", but the experiments were conducted on a single GPU.  It does not suffer from the multi-GPU, multi-node straggler problem that the paper uses as its primary motivation.

- The paper needs to quantify the overhead of LASER routing algorithm compared to top-k.

- Concerns on the claim of plug-and-play. Table 1 shows that hyper-parameters are set per-model, per-dataset, and are even "band-specific" (different values for early, middle, and final layers). This implies a "plug-and-tune" method that requires significant pre-analysis of prefill statistics  for every new task, undermining the ease-of-use.

**Questions:**

The "load-only" baseline is a trivial lower bound for balancing. Comparing against other simple inference-time heuristics would make this paper more solid. For example, a stochastic policy like "sample k experts from the top-c pool weighted by their scores"?

---

### Note · Authors · 2025-11-12

I have read and agree with the venue's withdrawal policy on behalf of myself and my co-authors.